# Micro-Gear Point Cloud Segmentation Based on Multi-Scale Point Transformer

Yizhou Su [1,2], Xunwei Wang [2,3], Guanghao Qi [1,2] and Baozhen Lei [1,2,*]

1 Beijing Key Laboratory of Information Service Engineering, Beijing Union University, Beijing 100101, China; empty0box@163.com (Y.S.); qigh9914@163.com (G.Q.)
2 College of Robotics, Beijing Union University, Beijing 100101, China; jdtxunwei@buu.edu.cn
3 Beijing Intelligent Machinery Innovation Design Service Engineering Technology Research Center, Beijing Union University, Beijing 100101, China
* Correspondence: leibaozhen@126.com; Tel.: +86-18911885860

**Abstract:** To address the challenges in industrial precision component detection posed by existing point cloud datasets, this research endeavors to amass and construct a point cloud dataset comprising 1101 models of miniature gears. The data collection and processing procedures are elaborated upon in detail. In response to the segmentation issues encountered in point clouds of small industrial components, a novel Point Transformer network incorporating a multiscale feature fusion strategy is proposed. This network extends the original Point Transformer architecture by integrating multiple global feature extraction modules and employing an upsampling module for contextual information fusion, thereby enhancing its modeling capabilities for intricate point cloud structures. The network is trained and tested on the self-constructed gear dataset, yielding promising results. Comparative analysis with the baseline Point Transformer network indicates a notable improvement of 1.1% in mean Intersection over Union (mIoU), substantiating the efficacy of the proposed approach. To further assess the method's effectiveness, several ablation experiments are designed, demonstrating that the introduced modules contribute to varying degrees of segmentation accuracy enhancement. Additionally, a comparative evaluation is conducted against various state-of-the-art point cloud segmentation networks, revealing the superior performance of the proposed methodology. This research not only aids in quality control, structural detection, and optimization of precision industrial components but also provides a scalable network architecture design paradigm for related point cloud processing tasks.

**Keywords:** point cloud segmentation; point cloud dataset; global feature; multi-scale fusion; micro-gear

## 1. Introduction

With the development of LiDAR, RGB-D cameras, and 3D structured light sensors, and the trend towards miniaturization, portability, and automation of acquisition devices becoming increasingly evident, the collection of point cloud data has become more convenient. Compared to two-dimensional data, point cloud data contain rich spatial structural information, making it a data type with a very high information density. Point cloud segmentation technology refers to the process of dividing a large amount of raw point cloud data into multiple subsets, each representing a separate entity or a relatively independent part within the depicted object. This technology is widely applied in fields such as autonomous driving [1–4], robotic perception [5–7], and 3D environment understanding [8–10].

Datasets serve as the foundation for deep learning. However, existing public datasets often focus on areas like autonomous driving and indoor/outdoor environments, with few datasets dedicated to the precision industrial component sector. Given that industrial parts often have complex shapes and small gears represent these components widely used across various sectors and that are irreplaceable in many aspects, studying such

representative industrial components can facilitate the broad application of deep learning in industrial production.

Based on this, the present study employs Metal Injection Molding [11–16] (MIM) technology and designs experiments based on factorial design [17], producing gears and using a 3D structured light scanner to collect point cloud data of all gear shapes throughout the MIM process, thereby constructing a dataset containing 1101 point cloud datasets of small gears. Subsequently, the gears are manually annotated and segmented.

Using the dataset created for this study, a multiscale feature fusion Point Transformer network is proposed, based on the PointTransformer [18] as the underlying network. This network incorporates multiple global feature extraction modules to extract features at multiple levels and employs an up-and-down sampling module [19] to integrate multiscale contextual information, enhancing the modeling capability for complex structured point clouds.

The structure of this paper is as follows. Section 2 introduces some commonly used public point cloud datasets and the MIM process along with the orthogonal experimental plan, followed by the point cloud data collection process, and finally the dataset's annotation and enhancement process. Section 3 introduces recent point cloud segmentation network models, details the network structure of PointTransformer, and elaborates on the Multilayer Feature Fusion Point Transformer (MFF-PT) network structure. Section 4 discusses experimental parameters, conducts ablation experiments to ensure the effectiveness of the model, compares it with other algorithms on the custom dataset, and presents the results. Finally, Section 5 summarizes the innovations and contributions of this research paper and points out its limitations.

## 2. A Dataset for Micro Metal Gears Based on MIM

In industrial applications, particularly in the processing of point clouds for specific industrial parts, professional point cloud datasets are extremely scarce. Precision components, such as small gears, feature complex geometric shapes and fine surface characteristics, which pose higher demands on point cloud processing algorithms. Consequently, the establishment of a dedicated industrial dataset for small gear point clouds becomes particularly crucial.

### 2.1. Related Work

Datasets play a crucial role in the field of point cloud deep learning. The training and validation of deep learning models rely on large, high-quality datasets. These datasets not only provide a rich array of samples for training models but also reflect the diversity of data found in the real world. Today, there are many high-quality point cloud datasets available, such as ModelNet [20], ShapeNet [21], and KITTI [1], as shown in Table 1 below, which have become cornerstones in the fields of computer vision and point cloud research. They cover a wide range of complexities, from simple objects to complex scenes, providing valuable resources for both the academic and industrial communities.

**Table 1.** Partial point cloud dataset.

| Datasets | Nature | Characteristics |
| --- | --- | --- |
| ShapeNet [21] | Virtual data | Clean, tidy, and labeled |
| ModelNet40 [20] | Virtual data | Clean, tidy, and labeled |
| KITTI [1] | Real data | Street scene, hollow, irregular, noisy |
| 3DMatch [22] | Real data | Indoor scenes, RGBD data, divided into training and testing sets |
| ASL Datasets Repository [1] | Real data | Architecture, terrain, irregularity, noise |
| Sydney Urban Objects Dataset [2] | Real data | Hollow, irregular, and noisy |
| The Stanford 3D Scanning Repository [3] | Real data | high quality |

[1] https://projects.asl.ethz.ch/datasets/ (accessed on 12 May 2024); [2] https://www.acfr.usyd.edu.au/papers/SydneyUrbanObjectsDataset.shtml (accessed on 12 May 2024); [3] http://graphics.stanford.edu/data/3Dscanrep/ (accessed on 12 May 2024).

### 2.2. Manufacturing of Micro Metal Gears through MIM

Metal Injection Molding (MIM) is an emerging powder metallurgy technology particularly well-suited for high-volume production of precision micro-sized components. The specific process flow diagram for this technique is depicted in Figure 1.

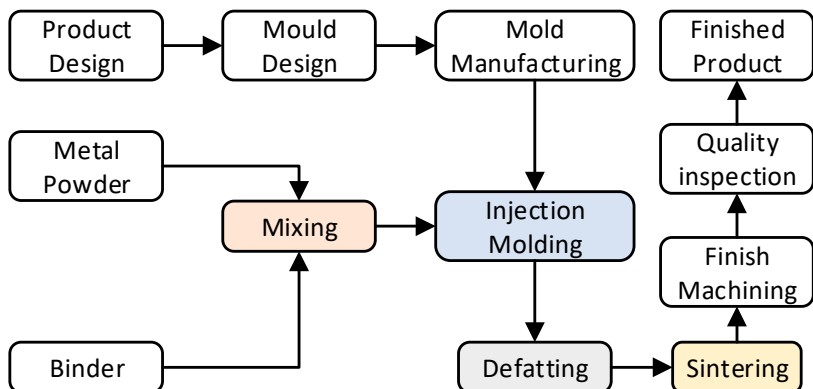

**Figure 1.** MIM process flowchart.

The Metal Injection Molding (MIM) process primarily consists of four key stages: mixing and granulating, molding, debinding, and sintering. In these stages, mixing and granulating is the first and a relatively independent step, which includes several sub-steps such as premixing of powder, preparation of binder, mixing of powder and binder, and finally the granulation process. The produced granules, referred to as feedstock, are used in the second stage of molding; during this molding stage, the feedstock is heated to a specific temperature to melt it, then injected under pressure into a prepared mold. The mold should have temperature controls to facilitate demolding and cooling, after which the parts cool down and are demolded. After this step, the parts are called green parts, at which point they already possess the complex structures required by the parts.

The third part, the debinding stage, is a unique step in the MIM process, where, due to the large amount of binder in the formed green parts, about 30% to 50% of the binder needs to be removed from the blank, which directly relates to the feedstock composition. After this step, the parts become extremely fragile; however, as the parts contain metal powder, they do not shrink or collapse.

The final stage, sintering, is crucial for the densification of the parts and differs from traditional powder metallurgy sintering. In traditional powder metallurgy, pressed parts already possess a relatively high density and only about 10% of porosity needs to be eliminated for densification. However, after the debinding process in MIM, up to about 40% porosity remains, leading to significant shrinkage during sintering.

Due to the significant changes in the shape of components throughout the MIM process, this study will collect point cloud data of the components at each stage of the MIM process to illustrate these unique transformations.

The experimental subject of the dataset is a spur gear with 17 teeth and a module of 0.5 mm, as illustrated in Figure 2. The specific parameters are outlined in Table 2.

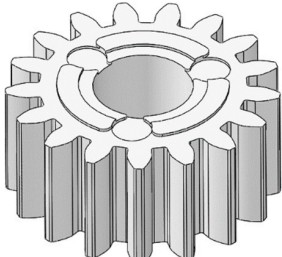

**Figure 2.** Geometric model of micro-gears.

**Table 2.** Gear size parameter information.

| Parameter Name | Parameter Value |
|---|---|
| Normal Module (mn/mm) | 0.5 |
| Number of Teeth (z) | 17 |
| Pressure Angle ($\alpha/°$) | 20 |
| Addendum Coefficient ($\chi$) | 0.06 |
| Pitch Circle Diameter (D/mm)) | 8.5 |
| Root Circle Diameter (Df/mm)) | 7.21 |
| Tip Circle Diameter (Da/mm)) | 9.56 |
| Tooth Pitch (pt/mm)) | 1.571 |

To fully reflect the characteristics of the MIM process, this study has designed and implemented an orthogonal experiment. There are four core process parameters in MIM: mold temperature, nozzle temperature, injection pressure, and holding time, which can affect the three-dimensional shape of the gears. The experimental scheme designed for this study is outlined in Table 3. The production experiments are conducted using the equipment shown in Figure 3.

**Table 3.** Experimental plan.

| Group | Mold Temperature (°C) | Nozzle Temperature (°C) | Injection Pressure (Bar) | Holding Pressure Time (s) |
|---|---|---|---|---|
| 1 | 80 | 185 | 70 | 0.2 |
| 2 | 80 | 190 | 80 | 0.5 |
| 3 | 80 | 197.5 | 100 | 1 |
| 4 | 80 | 205 | 120 | 1.5 |
| 5 | 80 | 210 | 130 | 1.8 |
| 6 | 90 | 185 | 80 | 1 |
| 7 | 90 | 190 | 100 | 1.5 |
| 8 | 90 | 197.5 | 120 | 1.8 |
| 9 | 90 | 205 | 130 | 0.2 |
| 10 | 90 | 210 | 70 | 0.5 |
| 11 | 105 | 185 | 100 | 1.8 |
| 12 | 105 | 190 | 120 | 0.2 |
| 13 | 105 | 197.5 | 130 | 0.5 |
| 14 | 105 | 205 | 70 | 1 |
| 15 | 105 | 210 | 80 | 1.5 |
| 16 | 120 | 185 | 120 | 0.5 |
| 17 | 120 | 190 | 130 | 1 |
| 18 | 120 | 197.5 | 70 | 1.5 |
| 19 | 120 | 205 | 80 | 1.8 |
| 20 | 120 | 210 | 100 | 0.2 |
| 21 | 130 | 185 | 130 | 1.5 |
| 22 | 130 | 190 | 70 | 1.8 |
| 23 | 130 | 197.5 | 80 | 0.2 |
| 24 | 130 | 205 | 100 | 0.5 |
| 25 | 130 | 210 | 120 | 1 |

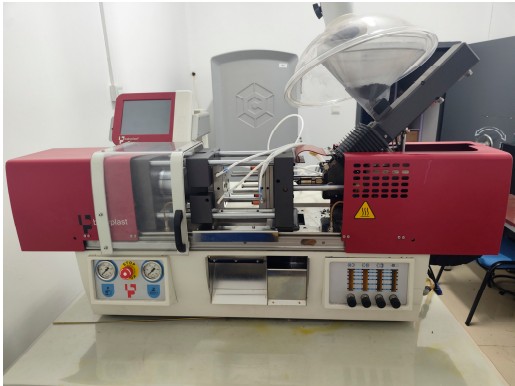

**Figure 3.** Babyplast6/10 Micro Injection Molding machine (The Rambaldi Group, Bologna, Italy).

Moreover, to further diversify the morphological characteristics of gears, various materials have been selected for gear manufacturing, including but not limited to popular materials such as 316L and 17-4ph. Additionally, experiments were conducted using each material with two different shrinkage ratios, contributing to the extended diversity of the dataset.

### 2.3. Collection and Processing of MIM Micro Metal Gear Dataset

Due to the need for comprehensive acquisition of gear surface data, an efficient, precise, safe batch inspection method is required. Traditional contact measurement techniques generally have a limit of a 0.2 module for measuring gear moduli, whereas non-contact measurement techniques based on diverse optical principles do not have this limitation.

Among 3D measurement technologies, the structured light projection method stands out for its high speed and precision. The core advantage of this measurement method lies in its non-contact nature and high degree of automation. It enables rapid and accurate measurements without touching the object's surface, significantly enhancing the efficiency and reliability of the measurements.

The German GOM ATOS Core three-dimensional optical scanning system, depicted in Figure 4, is employed for these measurements. The optical 3D scanning measurement system primarily consists of a high-precision stereo measurement head; an automated turntable; and the ATOS Professional 2018 measurement software. The equipment setup is illustrated in Figure 4. The point cloud data acquisition process is outlined in Figure 5.

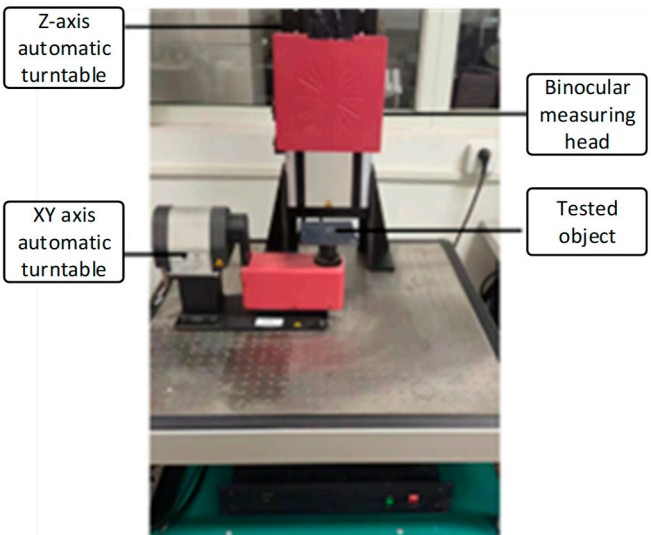

**Figure 4.** ATOS Core 3D optical scanning system.

It is important to note that before scanning, it is necessary to determine if the surface of the object being measured has any characteristics, such as reflectiveness, that could affect imaging. Surfaces that are too smooth require the application of a developer spray. In this experiment, the developer used is a mixture of 5% $TiO_2$ and 95% anhydrous ethanol, which is thoroughly mixed and then uniformly sprayed onto the surface of the component. The results are displayed in Figure 6.

Following the acquisition of raw point cloud data, a data refinement process is employed, involving trimming and removal of unnecessary planes and point cloud data to optimize the dataset. Subsequently, the ATOS software is utilized for polygonalization operations, enabling the generation of refined, non-overlapping triangular mesh data from the original point cloud data. The data before and after this process are illustrated in Figure 7. Such operations serve to further enrich the diversity of the dataset. Through experimental design and software processing, a total of 1101 high-quality point cloud data points were successfully collected.

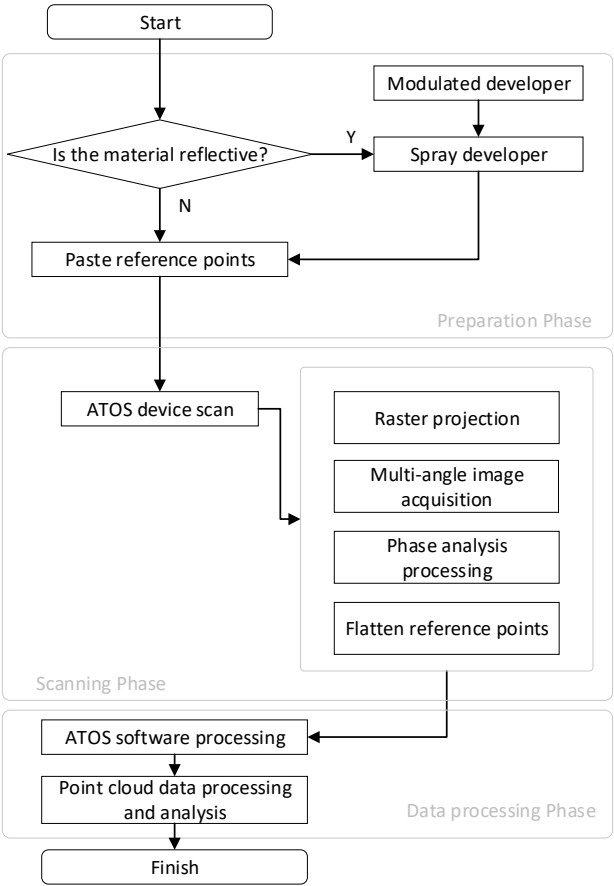

**Figure 5.** Point cloud data acquisition process.

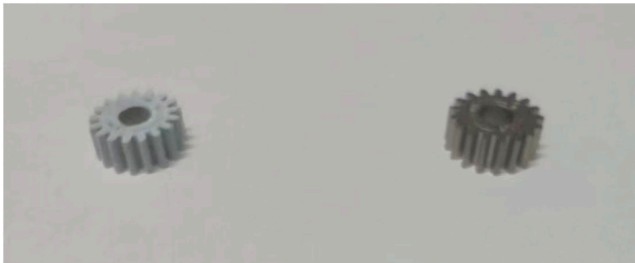

**Figure 6.** Comparison of gears before and after treatment with modifier.

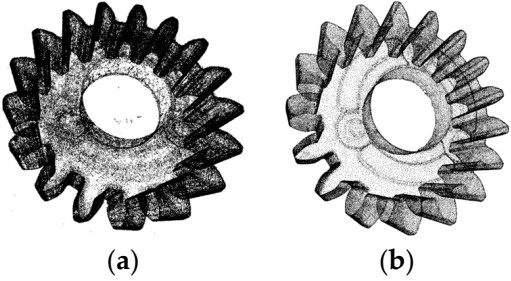

(**a**)                    (**b**)

**Figure 7.** Comparison of ATOS software optimization and processing of point cloud data. (**a**) Pre-processed point cloud data; (**b**) processed point cloud data.

### *2.4. PointTransformer Basic Principle*

Through an analysis of the distinctive features present in the point cloud data of gears, a rational approach is employed to partition the gear point cloud into four distinct components: the inner bore, teeth, end face, and noise, as illustrated in Figure 8.

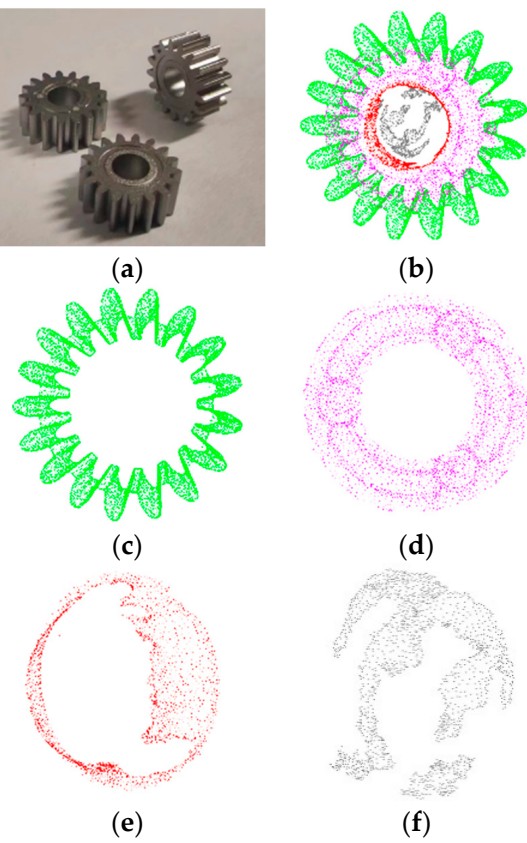

(**a**)      (**b**)

(**c**)      (**d**)

(**e**)      (**f**)

**Figure 8.** Self-built point cloud dataset. (**a**) Physical gear; (**b**) complete gear point cloud; (**c**) gear tooth point cloud; (**d**) point cloud of gear end face; (**e**) point cloud of gear inner hole; (**f**) data noise point cloud.

Due to the fact that each point cloud file typically encompasses over 50,000 points, and each point cloud dataset requires meticulous segmentation, the process becomes both intricate and time-consuming. The dataset comprises a total of 1101 distinct point cloud data points of miniature gears. The visual representation of the dataset is illustrated in Figure 9.

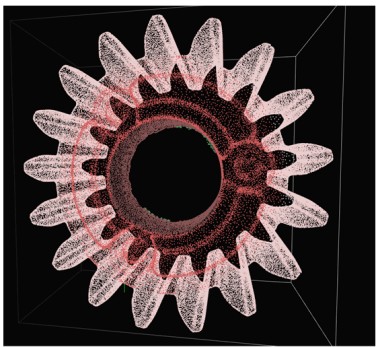

**Figure 9.** Tagged point cloud data.

## 3. Methods

Point cloud segmentation is fundamental to understanding point cloud data and holds significant value across multiple research and application domains. This section introduces a Multilayer Feature Fusion Point Transformer (MFF-PT) segmentation network, which aims to enhance the network's feature extraction capabilities to improve the accuracy of point cloud segmentation.

In the context of this research, which focuses on small gears, the precision requirements for different parts of these components vary considerably. For instance, the teeth and inner holes of a gear are far more critical than the gear's end face. If important parts can be segmented early in the process, it can effectively reduce subsequent computational load and minimize interference, making the segmentation of point clouds critically important.

### 3.1. Related Work

As hardware and deep learning technologies continue to evolve, numerous models for processing point cloud data have been proposed. However, the inherent characteristics of point cloud data, such as being unordered, irregular, and sparse, make its processing challenging. Techniques developed for 2D data processing cannot be directly applied to point cloud data due to these differences. Some network models attempt to convert point cloud data into more structured forms like voxels, clusters, or projections before performing feature extraction. Although this approach can effectively address some issues, it involves a preliminary transformation step, which tends to be resource-intensive and can lead to lower efficiency due to the high demands on memory and computational power.

Networks that process point cloud data directly also exist. Qi and others introduced PointNet [23], which was among the first to achieve end-to-end learning directly from point cloud data, using global pooling to directly handle the unordered nature of point clouds; PointNet++ [24] built on the foundation of PointNet by adding local region processing to capture more detailed local features. These two methods provide a basic structure for processing point cloud data, yet they still fall short in handling local details of point clouds; subsequent developments, such as PointCNN [25] and DensePoint [26], began to explore how to enhance performance by learning the local structures of point cloud data; to further improve model performance, researchers proposed several innovative methods. For example, GACNet [27] utilizes a graph attention mechanism to automatically learn dependencies between points, significantly enhancing the network's ability to parse complex spatial relationships. On the other hand, KPConv [8] performs convolutions directly on point clouds by defining learnable convolution kernels, handling multi-scale information, and providing a flexible method for processing point cloud data; later, PointTransformer [18] and PCT [28] introduced self-attention mechanisms [29], further enhancing the encoding of global dependencies in point clouds and optimizing the overall flow of information.

However, despite these advances bringing significant improvements, most networks still struggle with unclear connections between global and local features in point cloud data, leading to insufficient consideration of crucial information. Additionally, on specific applications such as custom datasets for small gears, these models often perform poorly, possibly due to not capturing fine local features adequately or lacking generalization capability under extreme conditions. For example, although HANet [30] and EPNet [31] excel in standard tasks, their applicability in specific small object recognition tasks remains limited, necessitating further consideration and integration of global and local information in the design of point cloud data processing networks.

Based on this, the present study builds upon the PointTransformer network, incorporating multiple global feature extraction modules to extract features at various levels. Additionally, an up-and-down sampling module is utilized, which integrates multiscale contextual information, thereby enhancing the modeling capabilities for complex structured point clouds.

### 3.2. PointTransformer Basic Principle

The Point Transformer (PT) is an early network utilizing self-attention mechanisms for classification and segmentation in point cloud data. As shown in Figure 10, its network structure combines the encoder–decoder architecture of the U-Net [32] network, skip connections, and self-attention mechanism layers (Point Transformer) to aggregate features of points within both global and local domains.

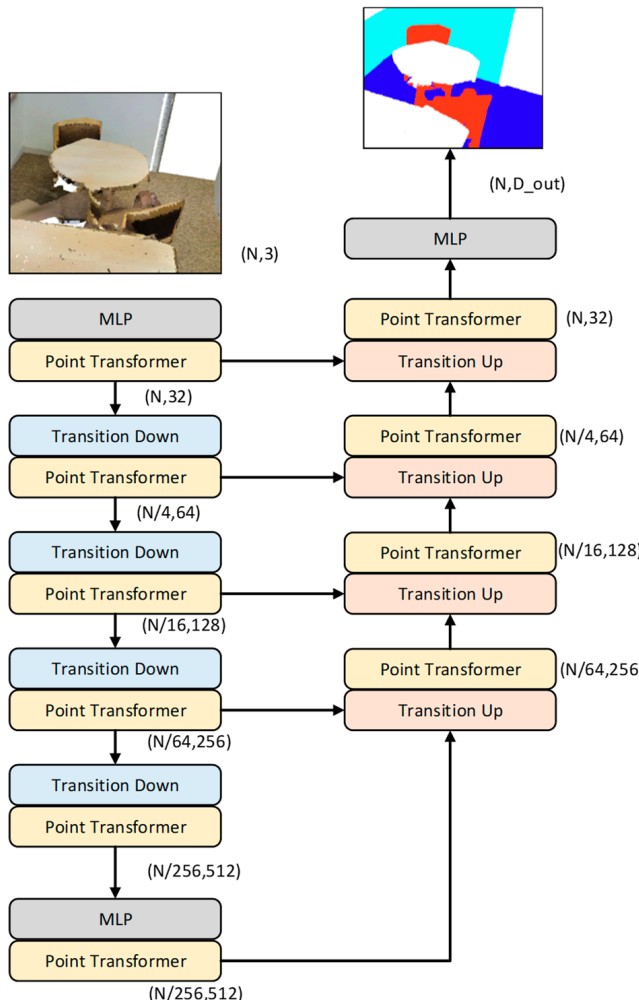

**Figure 10.** PointTransformer overall network structure diagram [13].

### 3.3. Multilayer Feature Fusion Point Transformer

Building upon the foundation of the Point Transformer, we introduce an enhancement by incorporating a global feature extraction network with multiscale fusion, resulting in the Multilayer Feature Fusion Point Transformer (MFF-PT). Additionally, an Up-Down-Up module is introduced to expand features, aiming to establish enhanced connections between global and local contextual relationships. The network architecture is depicted in Figure 11.

This network model preserves the encoder–decoder structure characteristic of the Point Transformer U-Net class. It comprises three main components: the Local Feature Encoder, the Global Feature Encoder, and the Decoder. The utilization of the Up-Down-Up module serves to facilitate improved integration of global and local contextual relationships.

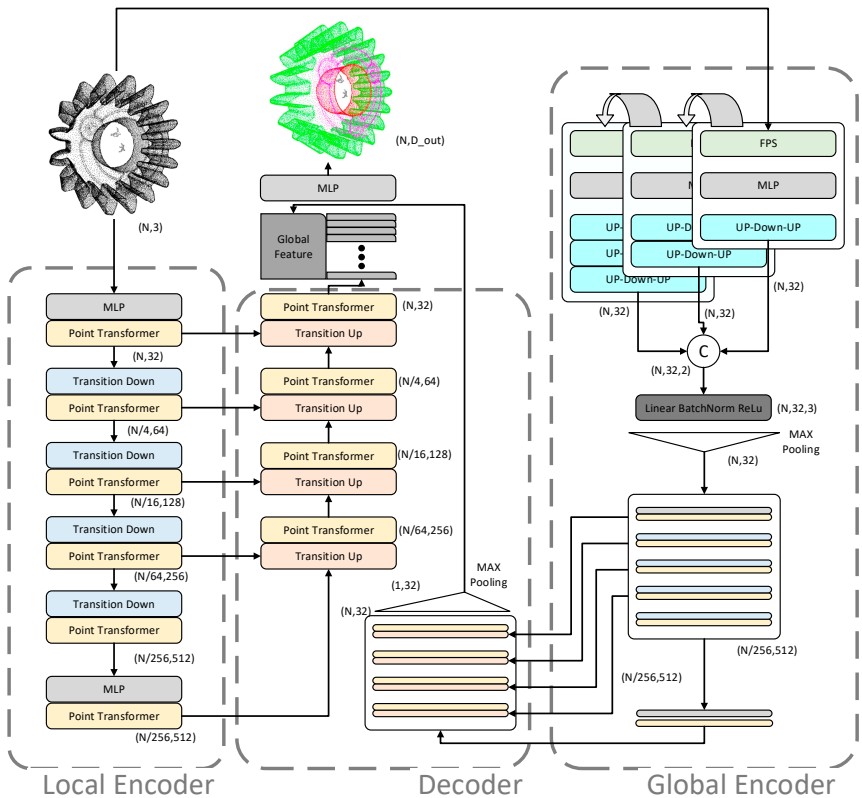

**Figure 11.** PointTransformer with multi-layer feature fusion.

### 3.3.1. Local Feature Encoder

The local feature encoder, essentially aligned with the encoder in the PointTransformer (PT), consists of a stack of Multilayer Perceptron (MLP) modules, Transition Down (TD) modules, and Point Transformer (PT) modules. This configuration is designed to extract local features from point cloud data and aggregate local information using a self-attention mechanism. The stacking of these modules allows the network to utilize a wider range of contextual information. Specifically, the module begins by expanding the point cloud features through the MLP, then employs the attention mechanism of the Point Transformer (PT) module to calculate the relationships between a point and several surrounding points. Following this, the Transition Down (TD) module performs downsampling, as depicted in Figure 12a. This process not only enriches the features but also reduces the number of points, facilitating subsequent operations such as upsampling and skip connections.

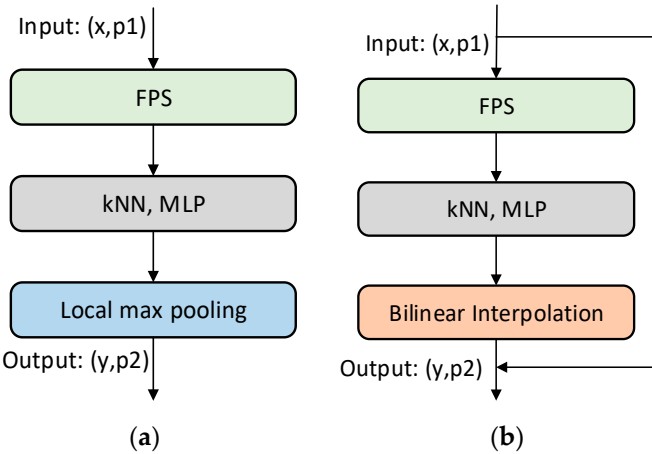

**Figure 12.** Up-and-down sampling modules. (**a**) Transition Down unit; (**b**) upsampling unit.

The PT module's structure and the Point Transformer layer schematic are displayed in Figure 13a. Comprising two linear layers and one PT layer, the module forms a PT block using residual connections, as illustrated in Figure 13b. The PT layer, central to the Encoder, calculates relationships among neighboring points within the local domain of each point, obtaining attention vectors. These vectors are then used to aggregate features of neighboring points:

$$y_i = \sum_{x_j \in x(i)} \rho\big(\gamma\big(\varphi(x_i) - \psi(x_j) + \delta\big)\big) \odot \big(\alpha(x_j) + \delta\big)\big) \tag{1}$$

$$\delta = \theta\big(P_i - P_j\big) \tag{2}$$

In the Point Transformer network, $y_i$ represents the output feature, and $\varphi$, $\psi$, and $\alpha$ are per-point feature transformations, such as MLPs. $\delta$ is a normalization function like softmax, and $\gamma$ is an MLP layer. Positional encoding, crucial for attention generation and feature transformation, is calculated using the three-dimensional coordinates $P_i$ and $P_j$ of two points, with $\theta$ being an MLP layer. The positional encoding function plays a key role in the attention mechanism, significantly impacting feature transformation.

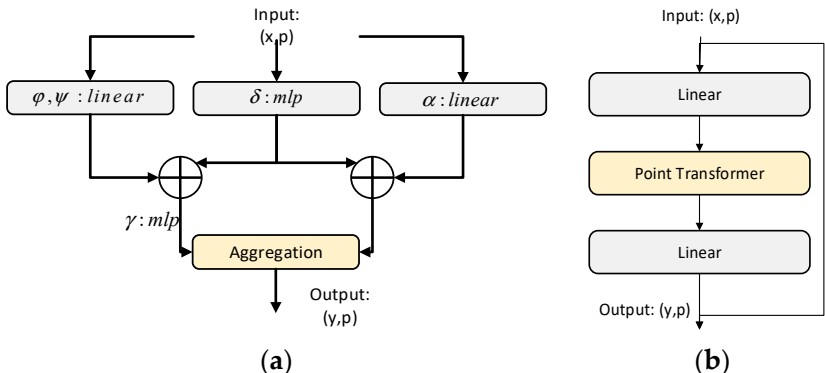

**Figure 13.** Point Transformer block structure. (**a**) Point transformer layer; (**b**) Point transformer block.

3.3.2. Global Feature Encoder

The Global Feature Encoder refines the number of points in the point cloud while preserving features, through multiple iterations of Farthest Point Sampling [33] (FPS). After each FPS, features are extracted using MLP. An Up-Down-Up (UDU) expansion unit enhances the point cloud features and generates new point data. The data, post-multiple layers of FPS and completion, have identical shapes. They are merged and processed through the LBR (Linear, Batch Normalization, ReLu) module, followed by Max Pooling for initial global feature extraction. The structure, similar to the Local Feature Encoder, includes a six-layer Encoder to further expand context and extract global features, as illustrated in Figure 11.

Figure 14 depicts the schematic of the Up-Down-Up (UDU) module, inspired by the point cloud completion network Pu-gan [19]. The UDU module initially uses MLP to enrich features, producing $F_1$. It then undergoes an upsampling and a downsampling process to obtain $F_2$ and $F_3$, respectively. Subsequently, the difference $\Delta_1$ between $F_1$ and $F_3$ is calculated. $\Delta_1$ is then upsampled to obtain $\Delta_2$. Finally, $\Delta_2$ and $F_2$ are merged to yield the final output. This method of upsampling effectively avoids overly complex training steps while efficiently increasing the number of effective point clouds.

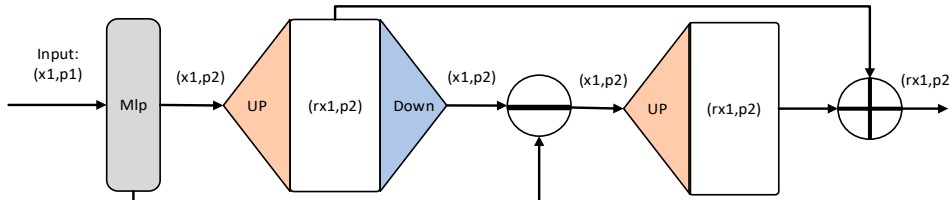

**Figure 14.** Up-Down-Up Point Cloud Expansion Unit.

### 3.3.3. Decoder

The Decoder, divided into two similar blocks, as depicted in Figure 11 is designed for extracting both global and local features. It consists of alternating Transition Up blocks and PT blocks, with the integration of Skip Connections. These connections link the Encoder's features to the downsampled features, recapturing details lost in the downsampling process. This procedure is repeated until the point cloud returns to its original form, a notable feature of the U-Net network. The Decoder adeptly fuses the point cloud's inherent information with the encoded context, methodically reconstructing and generating a high-quality, complete point cloud. Distinct from the Encoder, the Decoder includes additional Skip Connection attention layers, which are crucial for effective point cloud reconstruction. The Transition Up unit, as shown in Figure 15, leverages the Encoder's features for upsampling operations.

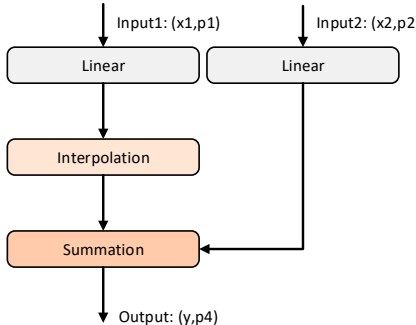

**Figure 15.** Upsampling unit.

After decoding through the Decoder, global features undergo another round of Max pooling for extraction. Following the approach of PointNet, these global features are concatenated to each local feature to facilitate point cloud segmentation operations, as illustrated in Figure 11. This method ensures that global context is effectively integrated with local details, enhancing the accuracy and effectiveness of the segmentation process.

## 4. Results and Discussion

### 4.1. Experimental Parameters

The experiments were conducted on a system equipped with an Intel i7-13700K CPU (Intel, Santa Clara, CA, USA), NVIDIA GeForce RTX 4090 GPU (NVIDIA, Santa Clara, CA, USA), running Microsoft Windows 11 Pro, with 16GB*2 DDR5 memory, using Pycharm as the integrated development environment. The CUDA version employed was 11.8, along with PyTorch version 1.9.0.

### 4.2. Point Cloud Component Segmentation Result Analysis

To verify the performance of the MFF-PT network in component segmentation, experiments were conducted using this network on the self-built dataset. The evaluation metric used was the mean Intersection over Union (mIoU), which calculates the IoU for each category and then takes the average of all category IoUs. The results are displayed in Figure 16.

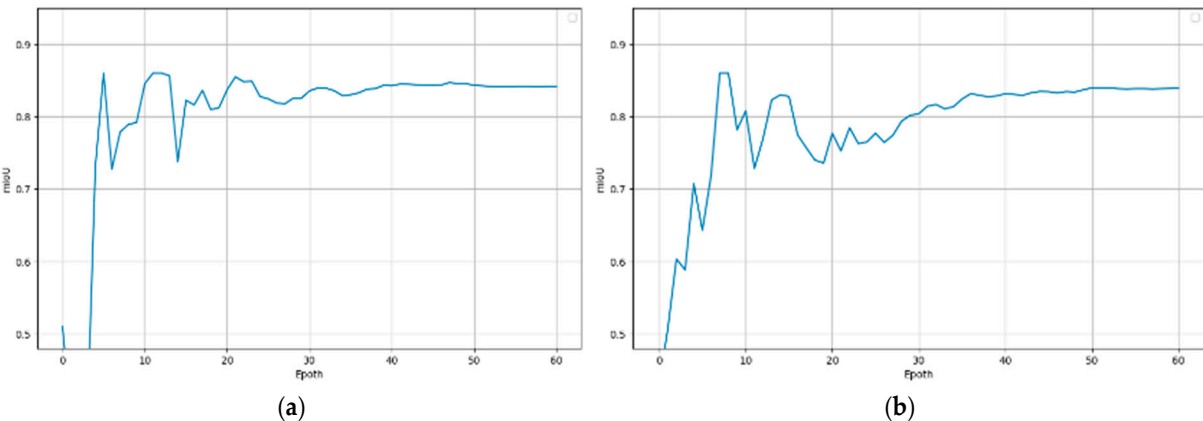

**Figure 16.** Training process performance indicators. (**a**) Multilayer Feature Fusion PointTransformer; (**b**) PointTransformer.

As depicted in the figure, after 600 training epochs, the original PT network began to converge around 300 epochs, while the improved MFF-PT network showed a convergence trend near 200 epochs. The MFF-PT network achieved an mIoU of 86.3%, compared to 85.2% for the PT network at the same number of epochs, marking an improvement of 1.1%. This indicates that the modified MFF-PT network has stronger learning capabilities on the custom micro-gear point cloud dataset than the original model.

### 4.3. Ablation Experiment

To further verify the effectiveness of the network, an ablation study was designed, as shown in Table 4. Since the MFF-PT, compared to the original PT network, primarily adds multiscale fusion for global feature extraction and an Up-Down-Up point cloud expansion module, the ablation study was organized into five groups.

Model A is the original PT model, serving as the control group.

Model B includes an added pathway of Farthest Point Sampling (FPS), MLP, and a stacked upsampling module (FMU). After FPS, the number of points is halved, but because the Down-Up-Down (DUD) module's comparison is for subsequent models (like Model D), another form of upsampling was necessary. Therefore, the upsampling method from the DUD module was used, which involves computing k-nearest neighbors and performing bilinear interpolation, as depicted in Figure 12b (upsampling unit). Due to having only one pathway of FMU, the max pooling operation is omitted.

Model C is similar to Model B but includes two FMU pathways, hence requiring the first max pooling operation.

Model D is essentially the same as Model C but includes three FMU modules.

Model E employs the DUD model, which corresponds to the MFF-PT model.

**Table 4.** Ablation experiment.

| Method | 1FMU | 2FMU | 3FMU | UDU | mIoU |
|--------|------|------|------|-----|------|
| A      |      |      |      |     | 85.2 |
| B      | √    |      |      |     | 85.3 |
| C      |      | √    |      |     | 85.5 |
| D      |      |      | √    |     | 85.7 |
| E      |      |      | √    | √   | 86.3 |

Table 4 shows that adding FMU modules indeed improves segmentation precision, but the degree of improvement is significantly related to the number of modules. A single FMU module increases mIoU by only 0.1%, possibly due to the absence of subsequent Max-Pooling, leading to poor global feature extraction. However, adding multiple FMU

modules can substantially enhance mIoU, with three modules achieving a 0.5% improvement. Replacing the upsampling module with the UDU module also contributes a 0.6% increase in the model's mIoU. This improvement is attributed to the UDU module's ability to effectively increase the number and features of newly generated point clouds through its multiple upsampling and downsampling processes.

### 4.4. Comparison with Existing Methods

To validate the effectiveness of the MFF-PT in comparison with classic networks, the study involved training and testing on a custom point cloud dataset using PointNet [23], PointNet++ [24], AGCN [34], DTNet [35], PRA-Net [36], PointTransformer [18], and FMM-PT, as illustrated in Table 5 and Figure 17. It was found that, generally, all networks performed worse on the custom dataset compared to their performance on the ShapeNet dataset. PT, PRA-Net, DTNet, and ours showed relatively better performance.

**Table 5.** Performance of various algorithms on self-built datasets.

| Method | mIoU |
| --- | --- |
| PointNet [23] | 83.4% |
| PointNet++(msg) [24] | 84.2% |
| AGCN [34] | 84.5% |
| DTNet [35] | 84.9% |
| PRA-Net [36] | 85.3% |
| PointTransformer [18] | 85.2% |
| Ours | 86.3% |

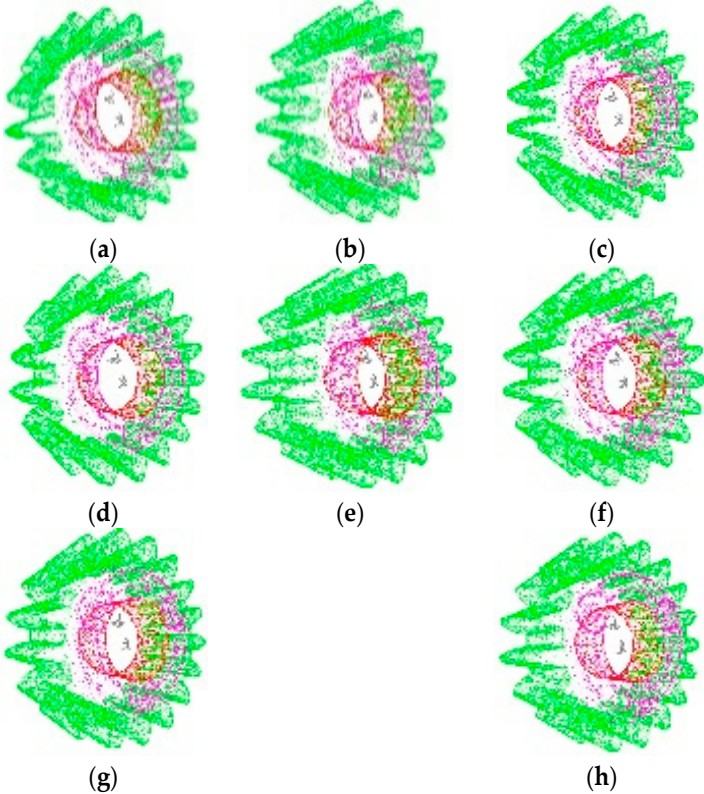

**Figure 17.** Visualization results of component segmentation. (**a**) Ground truth; (**b**) PointNet; (**c**) Point-Net++; (**d**) AGCN; (**e**) DTNet; (**f**) PRA-Net; (**g**) PointTransformer; (**h**) ours. Tip: In this picture, green represents teeth, pink represents gear end face, red represents inner hole, and gray represents noise.

To further validate the effectiveness of the network proposed in this study, multiple rounds of training and testing were conducted, with the network compared repeatedly against the next best performing networks, PRA-Net and Point Transformer. The results of these comparisons are shown in Figure 18. It was consistently found that MFF-PT performed better in each instance.

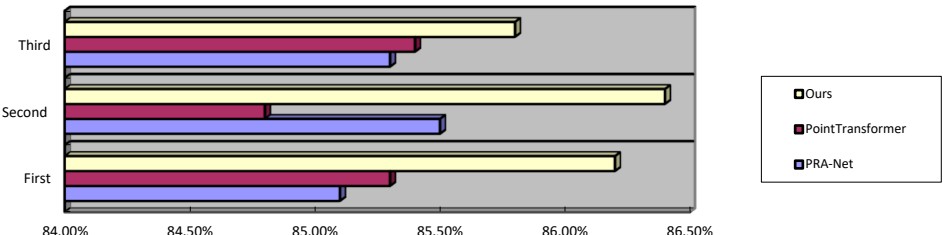

**Figure 18.** Multiple mIoU comparisons of PRA-Net, PointTransformer, and ours.

### 4.5. Limitations and Future Work

While the segmentation algorithm has shown improvements on the custom dataset, there remains room for further enhancement. For example, the use of a multiscale fusion structure has resulted in longer training times compared to the original network. Future plans include optimizing the downsampling methods and refining the Point Transformer layer structure to improve processing speed.

Additionally, the current limitation of having only one gear model due to the reliance on specific molds for Metal Injection Molding (MIM) hinders the generalizability of the model. New molds are already in production to manufacture two types of spiral bevel gears, aiming to enrich the dataset and enhance the model's robustness.

Plans are also in place to use the supplemented point cloud dataset for designing operations such as point cloud completion and registration. The goal is to optimize industrial production processes and parameters using deep learning methods, further integrating advanced technologies into the manufacturing sector to improve efficiency and product quality.

### 5. Conclusions

This study addresses the segmentation of point clouds for industrial small precision components by constructing a custom dataset containing 1101 small gear models. This dataset was obtained through a complete Metal Injection Molding (MIM) process and orthogonal experiments, ensuring it adequately reflects the characteristics of the MIM process.

Based on this dataset, a multiscale feature fusion Point Transformer network was proposed, which enhances the modeling capabilities for complex structured point clouds by incorporating global feature extraction modules and an upsampling module. Experimental results indicate that the proposed method achieves significant improvements over the original network, with evaluation metrics confirming its effectiveness and superiority.

The dataset and algorithm developed in this study are not only technical achievements but also lay a foundation for future research using deep learning point cloud data processing techniques in industrial production. However, both the dataset and the point cloud segmentation algorithm have limitations. The dataset should include more types of gears, and the segmentation algorithm should be further optimized to improve training efficiency and adapt to a broader range of gear point cloud data.

Looking forward, we plan to expand the dataset to accommodate all common types of gears and upgrade the segmentation algorithm by optimizing its modules to ensure more efficient training. Additionally, by fully utilizing the dataset, networks such as point cloud completion and registration will be designed, aiming to further promote the application of point cloud data in industrial production.

In summary, our dataset and segmentation algorithm have achieved significant progress, overcoming longstanding challenges and opening new avenues for the application of point cloud data. We will continue to advance research in this field.

**Author Contributions:** Conceptualization, Y.S. and G.Q.; methodology, Y.S. and X.W.; software, Y.S. and G.Q.; writing—original draft preparation, Y.S. and G.Q.; writing—review and editing, Y.S. and B.L.; supervision, X.W.; funding acquisition, B.L. All authors have read and agreed to the published version of the manuscript.

**Funding:** This paper is supported by a grant from the collaborative innovation project of Chaoyang District, Beijing (CYXC2102); is Supported by the Academic Research Projects of Beijing Union University (No. JZ10202003); is Supported by the National Natural Science Foundation of China (No. 5177050098).

**Institutional Review Board Statement:** Not applicable.

**Informed Consent Statement:** Not applicable.

**Data Availability Statement:** Restrictions apply to the availability of these data. Data were obtained from other research projects and are available from the corresponding authors with the permission of the respective owner.

**Conflicts of Interest:** The authors declare no conflicts of interest.

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
