# Peer review of "Micro-Gear Point Cloud Segmentation Based on Multi-Scale Point Transformer"

_applsci, doi:10.3390/app14104271_

Round 1

Reviewer 1 Report

Comments and Suggestions for Authors

The authors presented an interesting paper; despite this, the paper needs significant improvement. Below, I have included a few comments.

  • The paper needs better organization. Also, the authors need to improve its coherence. 
  • The paper could use more images to illustrate the comparison between the results obtained using the existing methods and the results obtained using the proposed method. 
  • Does the proposed method save time? If it does, show how much time it saves.
  • The paper needs a clear introduction section. The introduction method needs to contain more background information. Explain the paper's contribution to the body of knowledge. 
  • Additionally, the introduction contains the literature review. I suggest adding a literature review section. The authors have to check the literature review since it is incomplete. The paper needs to include important papers relevant to its topic.  
  • Line 102- 107. Move this paragraph to section 3.  
  • Figure 5. What is a spray developer?
  • Section 3 does not have any information. What goes here?
  • Table 4. What are the methods 1,2,3,4?
  • The paper does not have a conclusion section. Include the last paragraph in the conclusion section.

Author Response

Dear Expert,

First and foremost, I would like to express my heartfelt gratitude for the valuable feedback you provided during the review process.

After thoroughly reading your insightful comments, I have become acutely aware of the deficiencies in my paper. I have rewritten the section on related work and conducted several new rounds of experiments, which took some time. As this is my first journal submission, it is extremely important for my future. I hope to receive further guidance from you!

Additionally, I apologize for the delay in submitting the revised manuscript. This delay occurred as I encountered some unexpected difficulties while revising the paper and during sudden interviews. I am aware that this may have inconvenienced your work, and I deeply regret this.

Thank you once again for your understanding and support.

Sincerely,
Su Yizhou

Reviewer 2 Report

Comments and Suggestions for Authors

- The paper proposes a method for the segmentation of the point cloud of micro-gears by using neural networks. The problem has significance in quality control in industry. Although the addressed problem is very specific the paper can be relevant for a broader audience as well who are interested in deep learning techniques.

- The main question addressed by the research is to enhance the segmentation of the point cloud of micro-gears by using neural networks. The problem has significance in quality control in production of gears. Although the addressed problem is very specific the paper can be relevant for a broader audience as well who are interested in deep learning techniques.

- The paper focuses on detecting small-scale structures in industrial environment and it offers a more precise segmentation in this specific field than the other existing methods. 

- Point transformer networks are applied in the literature to point cloud segmentation. The proposed novel Multilayer Feature Fusion Point Transformer network enhances it with a global encoder module and upsampling module to provide multi-scale fusion.

- I consider the proposed method of point cloud segmentation novel and original. Point transformer networks are applied in the literature to point cloud segmentation. The proposed novel Multilayer Feature Fusion Point Transformer network enhances it with a global encoder module and upsampling module to provide multi-scale fusion.

- The main advantage of the proposed method is that it can achieve more precise segmentation for small-scale structures in industrial environment than the other existing methods.

- The applied methodology is appropriate to solve the addressed question. The proposed neural network is described in detail and illustrated with several figures to facilitate understanding its operation.

- Conclusion section is missing. The paper includes a paragraph on conclusions and future work at the end. Please add a section header before this paragraph.

- The main conclusion of the paper is that the proposed method outperforms the other existing methods, and that research contributes to the development of precision industrial component processing technologies. Several experiments were performed on a dataset consisting of more than 1000 point cloud data of miniature gears. The results support the conclusions of the paper although some more evaluation would be useful.The references are appropriate.

I have the following comments and recommendations to improve the paper:

- The proposed neural network is described in detail and illustrated with several figures to facilitate understanding its operation.

- Please describe how a point cloud segmentation can be used in quality control or production of micro-gears.

- If there are appropriate data, I recommend performing evaluation on applying the proposed method for fault detection.

- The running time of an algorithm is crucial in industry. Please provide a run time evaluation as well.

- Section 3. "Dataset and Experimental Parameters" is empty. I think it can be deleted because the dataset is described in Section 2.

- The paper includes a paragraph on conclusions and future work at the end. Please add a section header before this paragraph.

Please define all acronyms (e.g., MLP)

There are some typos:

Line 129: Principl -> Principle

Page 13, first paragraph: FMM -> MFF

Some labels on figures are hard to read. Please try to enlarge them.

Comments on the Quality of English Language

Only minor editing of English language required. It is easy to understand.

Author Response

(The authors gave the same response as above.)

Reviewer 3 Report

Comments and Suggestions for Authors

The article addresses an interesting issue, important from the point of view of the industry.

First of all, I would like to point out that my review concerns the already corrected version of the "v2" paper. Many changes have already been introduced in this version.

The substantive part is unquestionable.

However, I believe that the way the results are presented can be improved.

Some of the drawings are too small and therefore difficult to read. This is especially important in Chapter 4. Figure 19 should be enlarged so that the values on the axes and the axes captions are visible. I suggest enlarging the 8 graphics in Figure 20, they are illegible at their current scale.

Author Response

Dear Reviewer,

Thank you for your meticulous review and valuable guidance on my paper. Following your suggestion, I have enlarged the images in the paper to ensure they are clearer and easier to read. This paper is crucial for my successful graduation, and your guidance has been invaluable.

Thank you for your help and support!

Sincerely,

Yizhou Su
